# BEYOND ONE-SIZE-FITS-ALL: TAILORED BENCHMARKS FOR EFFICIENT MODEL EVALUATION

## ABSTRACT

Evaluating models on large benchmarks can be very resource-intensive, especially during a period of rapid model iteration. Existing efficient evaluation methods approximate the performance of target models by assessing them on a small static coreset derived from publicly available evaluation results of source models. However, these approaches rely on the assumption that each target model has a high prediction consistency with source models, which doesn't generalize well in practice, leading to inaccurate performance estimates. To fill this gap, we propose TAILOREDBENCH, a method that provides customized evaluations tailored to each target model. Specifically, a Global-coreset is first constructed as a probe to identify the most consistent source models for the target models with an adaptive source model selection strategy. Afterwards, a scalable K-Medoids clustering algorithm is proposed to extend the Global-coreset to tailored Native-coreset for each target model. According to the predictions on respective Native-coreset, we estimate the overall performance of target models with a calibrated restoration strategy. Comprehensive experiments on five benchmarks across over 300 models demonstrate that compared to best performing baselines, TAILOREDBENCH achieves an average reduction of 24.8% in MAE of accuracy estimates, showcasing strong effectiveness and generalizability.

## 1 INTRODUCTION

Scaling up models to larger size has led to remarkable advancements in their capabilities (Touvron et al., 2023; Ouyang et al., 2022), which also presents significant challenges for efficiently assessing them. For instance, Liang et al. (2022a) reported that evaluating a model with approximately 10 billion parameters on the HELM leaderboard can cost over \$1,700 via APIs or more than 1,200 GPU hours. Moreover, such overhead needs to be multiplied by $X$ when determining optimal training configurations during the development phase or selecting the best model and inference settings during the deployment phase among $X$ candidates.

To achieve efficient evaluation, some studies (Vivek et al., 2024; Polo et al., 2024) have explored the following paradigm: *step 1.* constructing example representations according to the predictions from a set of ***source models*** (which are freely available for many popular leaderboards[1,2,3]) ; *step 2.* clustering the whole benchmark and selecting the cluster centroids to form a coreset (typically no more than 100 examples); *step 3.* approximating the performance of ***target models*** under evaluation on the whole benchmark based on their predictions on the coreset. This paradigm is built upon the assumption that the predicted probability of the correct class is strongly correlated across models for many dozens of examples from a given benchmark. Thus, if the source models' predictions on example $a$ and $b$ exhibit a similar trend, then such similarity is expected to generalize to the target model — testing the target model solely on example $a$ is sufficient to predict its result on $b$.

Nevertheless, we find that such generalizability between source and target models does not necessarily hold. Following ANCHORPOINT (Vivek et al., 2024), we construct an $M$-dimensional representation vector based on the correctness of $M$ source models for each example and visualize them using t-SNE algorithm (Van der Maaten & Hinton, 2008). The closer two examples are, the more similar the source models perform on them (Figure 1a). Therefore, the cluster centroids (marked as stars) are selected to represent each cluster and form the coreset. However, when representing the examples using the predictions of target models (Figure 1b), the examples in the original coreset are no longer able to represent the corresponding clusters perfectly, indicating a moderate level of prediction consistency among source and target models. This lack of generalizability results in the performance of target models on the coreset failing to approximate their true performance accurately.

To address the aforementioned issue, we propose the TAILOREDBENCH method, which adaptively constructs model-specific evaluation coreset through a hybrid of priors and posteriors, enabling accurate and efficient evaluation. Specifically, we first construct a static G-set (Global-coreset) based on the metric of all the source models,

---

[1] https://huggingface.co/open-llm-leaderboard
[2] https://rank.opencompass.org.cn
[3] https://crfm.stanford.edu/helm

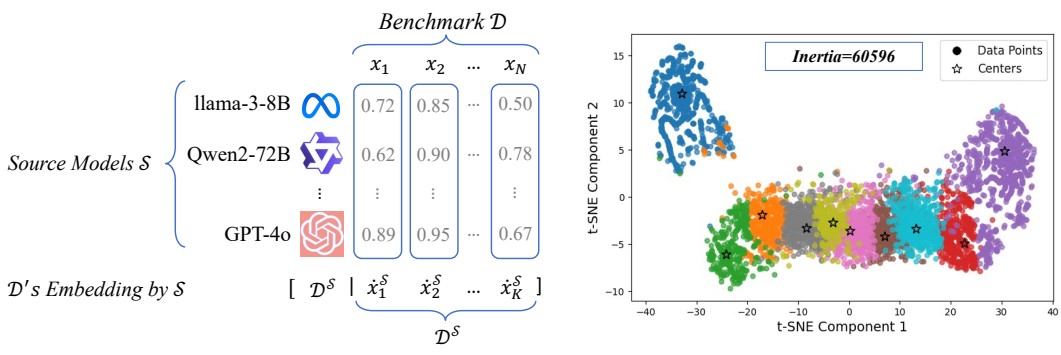

(a) Hellaswag under the representation of $\mathcal{D}^{\mathcal{S}}$.

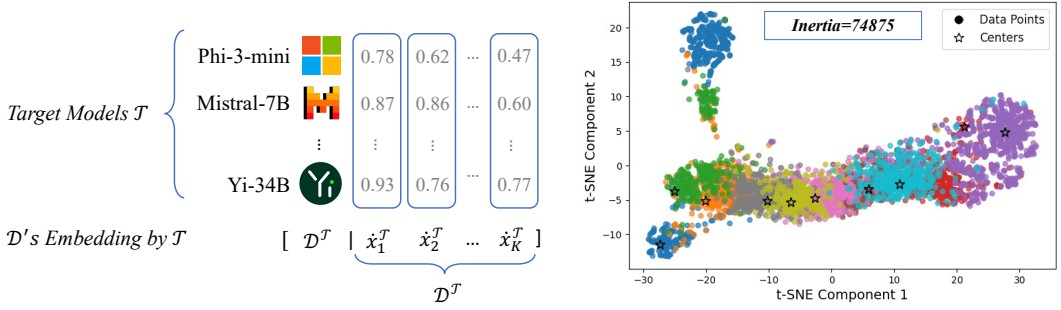

(b) Hellaswag under the representation of $\mathcal{D}^{\mathcal{T}}$.

Figure 1: The t-SNE visualization results of the Hellaswag benchmark under the metrics of source (above) and target (below) models' predictions. The larger inertia (sum of the sample distances to their cluster centers) below shows that the centroids obtained with the representations of source models are no longer capable to represent the whole benchmark for target models.

grounded on the prior that models generally share a moderate prediction consistency as discussed above. By applying an adaptive source model selection strategy, the predictions of target models on the G-set are used as a probe to select a Native Source Model set for each target model that has stronger prediction consistency with them. Based on this posterior, we design a scalable K-Medoids clustering technique to expand the G-set into an N-set (Native-coreset) for each target model, according to the representations under the metric of corresponding native source models. Finally, we approximate the overall performance of target models by employing a calibrated restoration strategy based on their predictions on the N-set.

We conduct extensive experiments on five benchmarks across more than 300 models, involving tasks in the fields of natural language and multi-modality. The experimental results demonstrate that compared to non-customized efficient evaluation baselines, TAILOREDBENCH can more accurately reflects the true performance of models (attain an average of 24.8% MAE degradation improvement on accuracy) under the same small-size inference budgets (generally 20~40 examples). Our contributions are summarized as follows:

1. We analyze that the existing efficient evaluation methods overestimate the prediction consistency across models, thus the static coreset they construct based on all the source models may fail to assess the target models accurately.
2. We propose TAILOREDBENCH method to conduct tailored evaluation on adaptively constructed N-set for each target model to attain more accurate evaluation results.
3. We conduct comprehensive experiments and analyses on multiple settings to validate the excellent effectiveness and strong generalizability of TAILOREDBENCH.

## 2 TAILOREDBENCH APPROACH

The TAILOREDBENCH method is founded on the principle of dynamically selecting the source models with highest prediction consistency and constructing N-set that can represent the whole benchmark for each target model. The core designs throughout the process are constructing the G-set (§2.2), selecting the native source models (§2.3), developing the N-set (§2.4), and estimating the overall performance (§2.5).

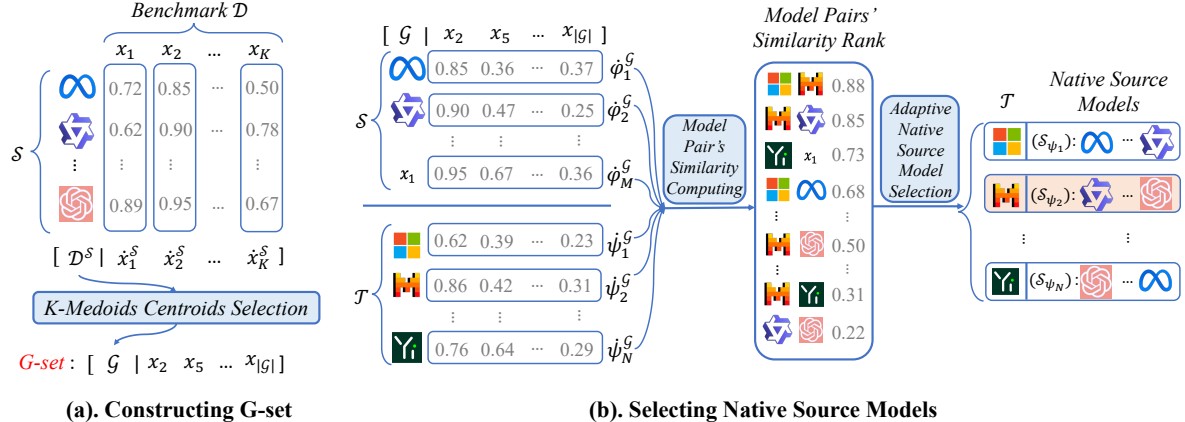

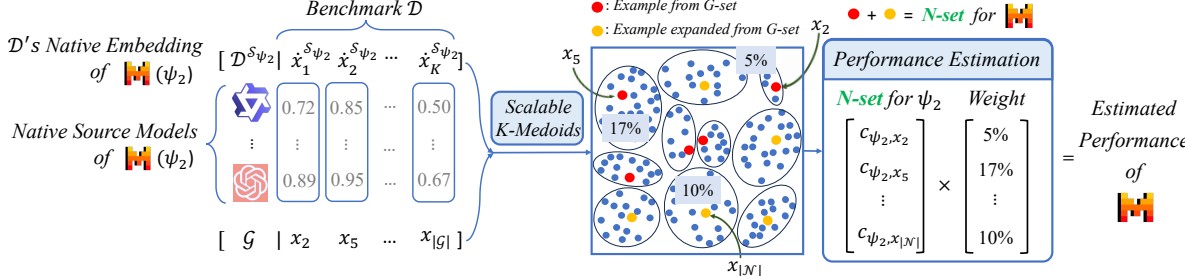

(c). Estimating the Performance of Target Model on N-set

Figure 2: Overview of TAILOREDBENCH.

## 2.1 TASK SET-UP

Let $\mathcal{D} = \{(x_k, y_k)\}_{k=1}^K$ represents a benchmark for evaluating, where $x_k$ is the input given to the model and $y_k$ is the expected output. We define the set of target models to be evaluated as $\mathcal{T} = \{\psi_m\}_{m=1}^M$, and a distinct set of source models as $\mathcal{S} = \{\varphi_n\}_{n=1}^N$, which we have access to their predictions across all examples in $\mathcal{D}$. Our goal is to accurately estimate the performance $\hat{P}_{\psi_m}$ of each target model $\psi_m$ and the ranking relationship across $\mathcal{T}$ with minimal model inference cost.

## 2.2 CONSTRUCTING G-SET

The G-set is designed as a probe for each target model to identify a set of source models with the highest prediction consistency. Thus, it is expected to be small yet representative of the benchmark and generalizable to any target models. To this end, we embed each example based on the correctness of its answers from every source model in the source model set, denoted as $\mathcal{D}^{\mathcal{S}} = \{\dot{x}_k^{\mathcal{S}}\}_{k=1}^K$, where the dot above means that $\dot{x}$ is a vector, and the superscript $\mathcal{S}$ indicates the model set that providing the correctness for each example, comprising the dimension of the example vector. The correctness used for encoding can either be a continuous value which reflecting the degree of model's correctness on each example (e.g., model's confidence) or a discrete binary value $\{0, 1\}$ indicating whether the model answered the question correctly, we will illustrate that our method naturally adapts to various formats of correctness in §3.1. Based on the $\mathcal{D}^{\mathcal{S}}$ we obtained, we employ K-Medoids clustering and the target function for obtaining G-set can be formulated as:

$$\min_{\mathcal{G}} \sum_{g=1}^{|\mathcal{G}|} \sum_{\dot{x}_k \in \mathcal{C}_g} \text{DIST}(\dot{x}_g^{\mathcal{S}}, \dot{x}_k^{\mathcal{S}})$$

$$\text{where } \dot{x}_g^{\mathcal{S}} = \left( [c_{\varphi_n, x_g}]_{n=1}^N \right)^\top = \begin{pmatrix} c_{\varphi_1, x_g} \\ c_{\varphi_2, x_g} \\ \vdots \\ c_{\varphi_N, x_g} \end{pmatrix}, \quad \dot{x}_k^{\mathcal{S}} = \left( [c_{\varphi_n, x_k}]_{n=1}^N \right)^\top = \begin{pmatrix} c_{\varphi_1, x_k} \\ c_{\varphi_2, x_k} \\ \vdots \\ c_{\varphi_N, x_k} \end{pmatrix} \quad (1)$$

In Eq. 1, $x_g$ is the example in $\mathcal{G} = \{x_g\}_{g=1}^{|\mathcal{G}|}$(G-set), $\mathcal{C}_g$ is the cluster whose centroid is $g$. $x_k$ represents the example belongs to $\mathcal{C}_g$, having stronger correlation with other examples from other clusters. $c_{\varphi_n, x_g}$ and $c_{\psi_n, x_k}$

represent the correctness of models $\varphi_n$ on example $x_g \in \mathcal{G}$ and $x_k \in \mathcal{D}$, respectively. DIST is the distance we use in clustering. In practice, we employ the Partitioning Around Medoids (PAM) algorithm (Kaufman & Rousseeuw, 2009), a typical K-medoids algorithm for the rapid and efficient optimization to solve this objective function.

## 2.3 ADAPTIVE NATIVE SOURCE MODEL SELECTION

To construct the Native Source Model Set $\mathcal{S}_{\psi_m}$ for each target model $\psi_m \in \mathcal{T}$, we propose an ADAPTIVE NATIVE SOURCE MODEL SELECTION strategy. This approach dynamically selects a subset of source models that exhibit the highest prediction consistency with the target model, ensuring that the subsequent tailored benchmark can accurately estimate the model performance.

To implement this strategy, we begin by obtaining the prediction results of both source and target models on the G-set $\mathcal{G}$, which serves as a posterior. For each source model $\varphi_n \in \mathcal{S}$ and target model $\psi_m \in \mathcal{T}$, we define their correctness vectors on the G-set as:

$$\dot{\varphi}_n^{\mathcal{G}} = \left( \left[ c_{\varphi_n, x_g} \right]_{g=1}^{|\mathcal{G}|} \right)^{\top} = \begin{pmatrix} c_{\varphi_n, x_1} \\ c_{\varphi_n, x_2} \\ \vdots \\ c_{\varphi_n, x_{|\mathcal{G}|}} \end{pmatrix}, \quad \dot{\psi}_m^{\mathcal{G}} = \left( \left[ c_{\psi_m, x_g} \right]_{g=1}^{|\mathcal{G}|} \right)^{\top} = \begin{pmatrix} c_{\psi_m, x_1} \\ c_{\psi_m, x_2} \\ \vdots \\ c_{\psi_m, x_{|\mathcal{G}|}} \end{pmatrix} \quad (2)$$

Here, the superscript $\mathcal{G}$ indicates that correctness is evaluated on the examples in the G-set, and each correctness vector captures the model's performance across all examples in $\mathcal{G}$. With the correctness vectors established, we proceed to determine the threshold for the number of Native Source Models by computing the distances between the correctness vectors of **all models** on the G-set:

$$d_{ij} = \text{DIST}(\dot{\phi}_i^{\mathcal{G}}, \dot{\phi}_j^{\mathcal{G}}) \quad (3)$$

where $i, j \in range(|\mathcal{S} \cup \mathcal{T}|)$ and $\phi$ represents any model. Next, we calculate the average prediction consistency $\bar{d}$ across all model pairs, which serves as a threshold for selecting source and target model pairs with high prediction consistency:

$$\bar{d} = \frac{2}{(N+M)(N+M-1)} \sum_{i<j} d_{ij} \quad (4)$$

where the summation is over all unique pairs $i < j$. Finally, we calculate the average number of source models whose consistency with target model exceeds threshold $\bar{d}$, denoted as $\bar{n}$, to be the optimal number of source models for the target models:

$$\bar{n} = \left\lfloor \frac{1}{M} |\{(\dot{\varphi}_n^{\mathcal{G}}, \dot{\psi}_m^{\mathcal{G}}) | \text{DIST}(\dot{\varphi}_n^{\mathcal{G}}, \dot{\psi}_m^{\mathcal{G}}) < \bar{d}\}| \right\rfloor \quad (5)$$

In this expression, $\bar{n}$ is adaptive and varies with different benchmarks and models, representing the near-optimal number of Native Source Models for each target model.

## 2.4 DEVELOPING N-SET

After selecting Native Source Models, we leverage the existing result of $\mathcal{S}_{\psi_m}$ to construct the most representative N-set for each target model $\psi_m$. Meanwhile, we hope to construct the N-set based on G-set so as to sufficiently utilize the target models' existing predictions on the G-set. However, current K-Medoids Clustering algorithms are inapplicable in this scalable scenario, for which we design a SCALABLE K-MEDOIDS CLUSTERING algorithm that can extend the G-set into the N-set.

For each target model $\psi_m$, the example $x_k \in \mathcal{D}$ is represented by a feature vector based on the correctness of the Native Source Models $\mathcal{S}_{\psi_m}$.

$$\dot{x}_k^{\mathcal{S}_{\psi_m}} = [c_{\varphi_1, x_k}, c_{\varphi_2, x_k}, \dots, c_{\varphi_{\bar{n}}, x_k}]^{\top} \quad (6)$$

Our Scalable K-Medoids Clustering algorithm proceeds as follows:

• **Initialization**: Fix the examples corresponding to the G-set $\mathcal{G}$ as part of the initial medoids. If the desired size of the N-set is $|\mathcal{N}|$ and the G-set contains $|\mathcal{G}|$ examples, then randomly select $|\mathcal{N}| - |\mathcal{G}|$ additional examples from $\mathcal{D} \setminus \mathcal{G}$ to complete the set of initial medoids.

• **Assignment Step**: Assign each example $x_k \in \mathcal{D}$ to the cluster $\mathcal{C}_\mu$ with the nearest medoid $x_\mu$:

$$x_k \in \mathcal{C}_\mu \text{ where } \mu = \arg\min_\mu \text{DIST}\left( x_\mu^{S_{\psi_m}}, x_k^{S_{\psi_m}} \right) \quad (7)$$

• **Medoid Update**: For each cluster $\mathcal{C}_\mu$ corresponding to a non-G-set medoid, update the medoid $x_\mu$ by selecting the example within $\mathcal{C}_\mu$ that minimizes the total distance to all other examples in the cluster, where the medoids

corresponding to G-set examples remain fixed and are not updated:

$$x_\mu = \arg \min_{x_i \in \mathcal{C}_\mu} \sum_{x_j \in \mathcal{C}_\mu \setminus x_i} \text{DIST}\left(x_i^{\mathcal{S}_{\psi_m}}, x_j^{\mathcal{S}_{\psi_m}}\right) \tag{8}$$

• **Convergence Check**: Repeat the assignment and medoid update steps until convergence, i.e., when the medoids no longer change or a maximum number of iterations is reached.

By incorporating the G-set into the medoids and fixing them during updates, we ensure that the G-set examples are integral to the clustering process, influencing the formation of clusters and the selection of additional N-set examples.

## 2.5 ESTIMATING TRUE PERFORMANCE

After constructing the N-set $\mathcal{N}_{\psi_m}$ for target model $\psi_m$, we evaluate $\psi_m$ on the examples in $\mathcal{N}_{\psi_m} \setminus \mathcal{G}$, since $\psi_m$ has already been evaluated on $\mathcal{G}$. The most straightforward way to estimate the performance of target model based on predictions on the N-set is to calculate the corresponding accuracy as an estimate of its overall performance. However, this method can overlook the size of the cluster that each example in the N-set represents. To this end, we apply a CALIBRATED RESTORATION strategy. Specifically, the performance $\hat{P}_{\psi_m}$ of $\psi_m$ on the entire benchmark $\mathcal{D}$ is estimated by weighting the performance on the N-set according to the proportion of examples represented by each cluster:

$$\hat{P}_{\psi_m} = \sum_{\mu=1}^{|\mathcal{N}|} \left( \frac{|\mathcal{C}_\mu|}{|\mathcal{D}|} \cdot c_{\psi_m, x_\mu} \right) \tag{9}$$

In this way, the cluster size bias can be well calibrated and $\hat{P}_{\psi_m}$ can be seen as an unbiased estimate of target model $\psi_m$.

## 2.6 THE ELEMENT-WISE DISTANCE

Previous work employs correlation distance to estimate the consistency between examples, operating under the assumption that scoring patterns between models or examples exhibit linear trends. However, this approach can encounter limitations, particularly when attempting to establish linear relationships between discrete numerical embeddings across different models, often resulting in a significant decline in performance. In contrast, the TAILOREDBENCH utilizes element-wise distance metrics, such as Manhattan distance and cosine similarity, to effectively capture individual discrepancies in correctness vectors without relying on a strict linear relationship. Our experiments in Table 7 demonstrate that element-wise metrics outperform correlation distance in constructing representative G-sets and N-sets, as well as in selecting Native Source Models. Consequently, we use Manhattan distance as the primary DIST metric in the TAILOREDBENCH method.

# 3 EXPERIMENTS

## 3.1 EXPERIMENTAL SETUP

**Benchmarks and Models**  To evaluate the effectiveness and generalizability of TAILOREDBENCH, we conduct experiments on five diverse benchmarks, involving tasks in the fields of natural language and multi-modality:

- **ARC-Challenge** (Clark et al., 2018): Consists of 1,172 grade-school science questions requiring scientific reasoning. We collect the outputs of 153 models on this benchmark.
- **HellaSwag** (Zellers et al., 2019): Contains over 10,000 validation examples for commonsense inference. For computational efficiency, we use the first 6,000 examples and gather the outputs of 139 models on this subset.
- **GSM8K** (Cobbe et al., 2021): Includes 1,319 grade-school math problems designed to test mathematical reasoning. We summarize the evaluation results of 150 models on this benchmark.
- **Winogrande** (Sakaguchi et al., 2021): Comprises 1,267 examples presenting challenging pronoun resolution tasks in commonsense reasoning. We summarize the evaluation results of 150 models on this dataset.
- **POPE** (Li et al., 2023): Contains 5,127 examples aimed at evaluating hallucination in multimodal models. We summarize the evaluation results of 99 models on this benchmark.

In our method, the correctness for each model can be either the output or the evaluation result on each specific example, represented as either continuous scores or discrete values. For *ARC-Challenge* and *HellaSwag* benchmark, we utilize continuous values representing the models' confidence in the correct option. For *GSM8K*, *Winogrande*,

Table 1: Results on ARC-Challenge benchmark. Values in bold underlined represent the best and the second-best results respectively. If the bold value is not marked with ∗, it signifies that it is statistically significantly better than the second-best value ($p - \text{value} < 0.05$). This applies to all tables below.

| Inference Count | 20 | | 25 | | 30 | | 35 | | 40 | |
|---|---|---|---|---|---|---|---|---|---|---|
| | $\tau \uparrow$ | MAE↓ | $\tau \uparrow$ | MAE↓ | $\tau \uparrow$ | MAE↓ | $\tau \uparrow$ | MAE↓ | $\tau \uparrow$ | MAE↓ |
| RANDOM | 0.626 | 0.078 | 0.659 | 0.065 | 0.676 | 0.067 | 0.694 | 0.062 | 0.712 | 0.057 |
| ANCHOR POINTS | 0.662 | 0.064 | 0.663 | 0.058 | 0.676 | 0.053 | 0.713 | 0.048 | 0.714 | 0.043 |
| GP-IRT | 0.589 | 0.046 | 0.620 | 0.046 | 0.662 | 0.036 | 0.681 | 0.036 | 0.695 | 0.029 |
| TAILOREDBENCH | **0.705** | **0.029** | **0.731** | **0.027** | **0.748** | **0.026** | **0.756** | **0.025** | **0.767** | **0.024** |

Table 2: Results on Hellaswag benchmark.

| Inference Count | 20 | | 25 | | 30 | | 35 | | 40 | |
|---|---|---|---|---|---|---|---|---|---|---|
| | $\tau \uparrow$ | MAE↓ | $\tau \uparrow$ | MAE↓ | $\tau \uparrow$ | MAE↓ | $\tau \uparrow$ | MAE↓ | $\tau \uparrow$ | MAE↓ |
| RANDOM | 0.811 | 0.083 | 0.836 | 0.077 | 0.850 | 0.066 | 0.863 | 0.060 | 0.871 | 0.058 |
| ANCHOR POINTS | 0.860 | 0.060 | 0.880 | 0.061 | 0.877 | 0.067 | 0.897 | 0.059 | 0.898 | 0.057 |
| GP-IRT | 0.724 | 0.062 | 0.776 | 0.053 | 0.810 | 0.043 | 0.827 | 0.038 | 0.849 | 0.032 |
| TAILOREDBENCH | **0.898** | **0.020** | **0.906** | **0.018** | **0.910** | **0.017** | **0.911** | **0.017** | **0.916** | **0.016** |

Table 3: Results on GSM8k benchmark.

| Inference Count | 20 | | 25 | | 30 | | 35 | | 40 | |
|---|---|---|---|---|---|---|---|---|---|---|
| | $\tau \uparrow$ | MAE↓ | $\tau \uparrow$ | MAE↓ | $\tau \uparrow$ | MAE↓ | $\tau \uparrow$ | MAE↓ | $\tau \uparrow$ | MAE↓ |
| RANDOM | 0.811 | 0.062 | 0.828 | 0.055 | 0.839 | 0.052 | 0.847 | 0.049 | 0.858 | 0.044 |
| ANCHOR POINTS | 0.786 | 0.087 | 0.791 | 0.079 | 0.796 | 0.073 | 0.800 | 0.071 | 0.799 | 0.071 |
| GP-IRT | 0.787 | 0.055 | 0.807 | 0.047 | 0.829 | 0.041 | 0.842 | 0.038 | 0.858 | 0.034 |
| TAILOREDBENCH | **0.851** | **0.041** | **0.853** | **0.039** | **0.862** | **0.036** | **0.866** | **0.035** | **0.879** | **0.032** |

Table 4: Results on Winogrande benchmark.

| Inference Count | 20 | | 25 | | 30 | | 35 | | 40 | |
|---|---|---|---|---|---|---|---|---|---|---|
| | $\tau \uparrow$ | MAE↓ | $\tau \uparrow$ | MAE↓ | $\tau \uparrow$ | MAE↓ | $\tau \uparrow$ | MAE↓ | $\tau \uparrow$ | MAE↓ |
| RANDOM | 0.373 | 0.078 | 0.408 | 0.067 | 0.446 | 0.062 | 0.470 | 0.055 | 0.492 | 0.052 |
| ANCHOR POINTS | 0.472 | 0.086 | 0.487 | 0.085 | 0.514 | 0.075 | 0.521 | 0.087 | 0.518 | 0.073 |
| GP-IRT | 0.263 | 0.041 | 0.313 | 0.038 | 0.353 | 0.038 | 0.392 | 0.036 | 0.419 | 0.034 |
| TAILOREDBENCH | **0.543** | **0.032** | **0.561** | **0.031** | **0.588** | **0.028** | **0.588** | **0.027** | **0.605** | **0.025** |

Table 5: Results on POPE benchmark.

| Inference Count | 20 | | 25 | | 30 | | 35 | | 40 | |
|---|---|---|---|---|---|---|---|---|---|---|
| | $\tau \uparrow$ | MAE↓ | $\tau \uparrow$ | MAE↓ | $\tau \uparrow$ | MAE↓ | $\tau \uparrow$ | MAE↓ | $\tau \uparrow$ | MAE↓ |
| RANDOM | 0.488 | 0.058 | 0.510 | 0.054 | 0.507 | 0.048 | 0.515 | 0.044 | 0.547 | 0.040 |
| ANCHOR POINTS | 0.474 | 0.040 | 0.483 | 0.038 | 0.518 | **0.034** | **0.547** | **0.033** | **0.556** | **0.031** |
| GP-IRT | 0.481 | **0.038** | 0.470 | **0.037** | 0.462 | 0.036 | 0.482 | 0.034 | 0.477 | 0.033 |
| TAILOREDBENCH | **0.509** | 0.043 | **0.526** | 0.041 | **0.531**∗ | 0.043 | **0.547**∗ | 0.038 | 0.551 | 0.039 |

and *POPE* benchmark, we use discrete values in {0, 1} as the correctness, where 1 indicates a correct answer and 0 indicates an incorrect answer.

All models evaluated on ARC-Challenge, HellaSwag, GSM8K, and Winogrande are sourced from the Open LLM Leaderboard (Beeching et al., 2023). The models and data for the POPE benchmark are obtained from the Open-Compass Leaderboard (Contributors, 2023). For each benchmark, we randomly and evenly partition the models into source and target model sets.

**Baseline and Evaluation Metrics**   We compare our TAILOREDBENCH method against three baselines: *Random Sampling*, *Anchor Points* (Vivek et al., 2024), and *gp-IRT* (Polo et al., 2024). The *Random Sampling* randomly selects a subset of examples from the benchmark to estimate model performance, serving as a basic reference point. *Anchor Points* uses K-Medoids to select a fixed set of representative examples based on the source models to estimate the performance of target models. *gp-IRT* method employs an Item Response Theory model trained on the predictions of the source models to estimate target models' performance on the full benchmark. Similar to our approach, we train the gp-IRT model using the predictions of the same set of source models and estimate the performance of the same set of target models. We evaluate the performance of these methods with two metrics: *Kendall's $\tau$ Correlation Coefficient* and *Mean Absolute Error (MAE)*. Kendall's $\tau$ measures the ordinal association between the estimated and true model rankings, effectively capturing how well our method preserves the relative performance order of the target models. MAE quantifies the average absolute difference between the estimated and true performance scores, assessing the precision of our method in estimating individual model performances.

## 3.2 EXPERIMENT ANALYSES

**TailoredBench: Effective Ranking and Estimation of Model Performances.**   Tables 1 to 5 present a comprehensive comparison between our TAILOREDBENCH method and baseline approaches across all benchmarks. Among these tables, bolded results indicate the best performance in each column, while underlined results represent the second-best performance. In our experiments, we allocated 10 examples to the G-set and averaged the outcomes over 100 randomized trials to ensure statistical reliability. The objective is to evaluate and compare the performance of our method against established baseline methods across varying inference counts. Specifically, the inference count—defined as the number of examples in the N-set for our method—is varied between 20 and 40. As presented in the accompanying tables, within the inference count range of 20 to 40, our method consistently outperforms the baseline approaches based on Kendall's $\tau$ metric. The improvement highlights our method's superior capability in accurately reconstructing the ranking of target models' capabilities using a limited amount of evaluation data.

Furthermore, our approach demonstrates consistent superiority over the baselines in terms of the Mean Absolute Error (MAE) metric across all benchmarks. On average, our method achieves a 24.8% reduction in estimation MAE loss, indicating a more precise estimation of the target models' actual performance. Notably, compared to the static AnchorPoints method, our MAE is typically reduced by half. This underscores our method's effectiveness in identifying a more representative N-set for each target model, thereby enhancing estimation accuracy. We further calculated the accuracy of our method in ranking the performance between every pair of target models. The results show that the accuracy reached 95.9% on the Hellaswag benchmark and 92.8% on the GSM8K benchmark.

The variance comparison between our method and the baselines is detailed in Appendix A, revealing that our method exhibits significantly lower variance. This lower variance signifies a higher level of robustness and consistency in our method's performance compared to the baseline approaches.

Moreover, results marked with an $*$ in our tables denote instances where a one-sided Z-test is conducted on the outcomes of 100 repeated experiments using our method versus 100 experiments with the baseline method. In these cases, the p-value for the hypothesis that our method's results are greater than those of the baseline exceeds 0.05. This indicates that, in these instances, we cannot statistically confirm that our method outperforms the baseline.

**Impact of Native Source Model Selection on Method Performance.**   In this section, we analyze the impact of the number of Native source models and the prediction consistency between the Native source models and the target models on our method. In the left panel of Figure 3, we maintain the overall prediction consistency between the Native source models and the target models constant, while varying the proportion of the source models designated as Native Source Models from 20% to 100% for the target models. It is

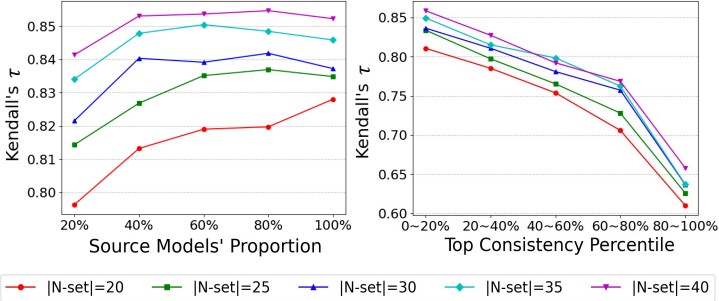

Figure 3: Analyses of Native Source Model Quantity and Prediction Consistency on GSM8k Benchmark.

evident that when the inference count ranges from 20 to 40, the performance of our method progressively improves with the increase in the number of Native Source Models. This is because, as the feature length increases, there is a greater possibility of reducing the noise impact caused by the prediction inconsistency between the source and target models. Dually, in the right panel of Figure 3, we demonstrate the performance of our method with a fixed number of Native Source Models, selecting those whose prediction consistency is ranked in the top 20%, 20%~40%, up to 80%~100% to the target model. The horizontal axis in this figure represents the Top Consistency Percentile, corresponding to these prediction consistency ranking ranges. It can be observed that across all inference counts, the performance decreases as the prediction consistency between the Native Source Models and the target model diminishes. Additional results on more benchmarks are provided in Appendix B and C. In summary, both the number of native source models and their prediction consistency with the target model significantly impact the performance of the method. This underscores the necessity of selecting source models that exhibit the highest prediction consistency with the target model while balancing the effects of the number of selected source models. Subsequently, we will demonstrate that TAILOREDBENCH can adaptively select an optimal set of Native Source Models for each target model, achieving near-optimal results.

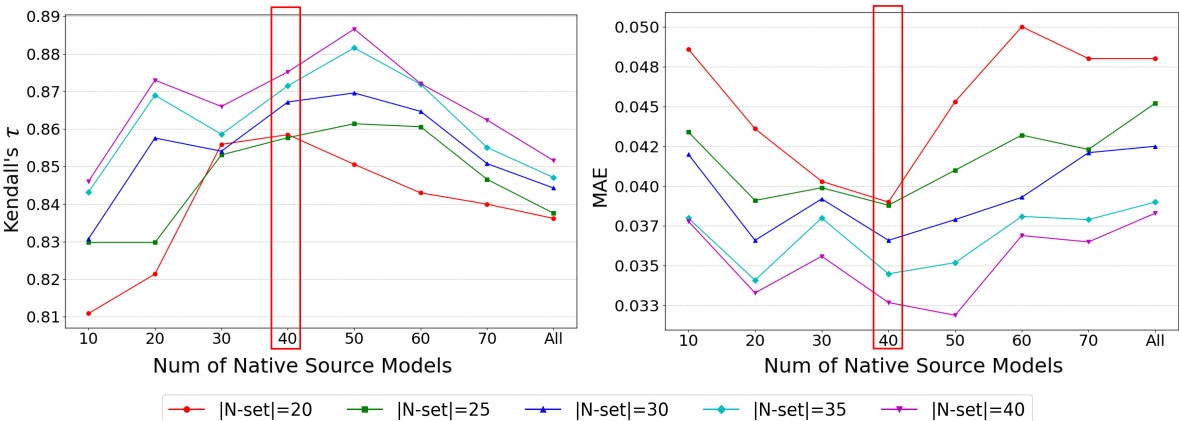

Figure 4: Effect of the Number of Native Source Models in Our Method on GSM8k Benchmark.

**TailoredBench Method Adaptively Selects Optimal Native Source Model Sets.** Figure 4 illustrates the performance of our method on the GSM8k benchmark when selecting Source Models with the top-k prediction consistency to the target model as the Native Source Models. The results show that Kendall's $\tau$ coefficient initially increases and then decreases as the number of Native Source Models grows, while the MAE shows an initial decrease followed by an increase. This phenomenon corresponds to what we observed in Figure 3. Specifically, when only a small number of Native Source Models are selected, despite their high prediction consistency to the target model, the limited quantity introduces more noise during the clustering of benchmarks constructed from embeddings provided by these models, thereby reducing clustering performance. Increasing the number of selected Native Source Models alleviates this issue and enhances performance until an optimal point is reached.

However, selecting too many Native Source Models gradually incorporates models with lower prediction consistency to the target model, which degrades the method's effectiveness. Our method effectively addresses this problem by adaptively selecting the near-optimal number of Native Source Models for all benchmarks. For instance, as shown in Figure 4, our approach selects the 40 Native Source Models for each target model on the GSM8k benchmark, achieving nearly optimal performance. Further experiments pertaining to this section are detailed in Appendix D.

Table 6: Impact of G-set Size on Performance Across Benchmarks.

| $|$G-set$|$ | ARC-Challenge | | Hellaswag | | GSM8k | | Winogrande | | POPE | |
|---|---|---|---|---|---|---|---|---|---|---|
| | $\tau \uparrow$ | MAE $\downarrow$ | $\tau \uparrow$ | MAE $\downarrow$ | $\tau \uparrow$ | MAE $\downarrow$ | $\tau \uparrow$ | MAE $\downarrow$ | $\tau \uparrow$ | MAE $\downarrow$ |
| 5 | 0.711 | 0.029 | 0.912 | 0.018 | 0.842 | 0.042 | 0.613 | 0.031 | 0.534 | 0.044 |
| 10 | **0.748** | 0.027 | 0.910 | 0.017 | **0.862** | **0.036** | 0.588 | **0.028** | 0.531 | 0.043 |
| 15 | 0.747 | **0.026** | **0.913** | **0.016** | 0.845 | 0.041 | 0.598 | 0.029 | **0.546** | 0.041 |
| 20 | 0.740 | **0.026** | **0.913** | 0.017 | 0.842 | 0.042 | 0.610 | 0.029 | 0.529 | **0.039** |
| 25 | 0.722 | 0.027 | 0.908 | 0.018 | 0.851 | 0.040 | **0.639** | 0.030 | 0.533 | 0.042 |

**10 Examples are Sufficient for the Probe.** We conducted an ablation study to examine the impact of the number of examples in the G-set on our method's performance. We fixed the number of examples in the N-set at 30 and varied the number of examples in the G-set from 5 to 25, evaluating across all benchmarks. The experimental results are presented in Table 6. We observe that, overall, as the number of examples in the G-set increases, the Kendall's $\tau$ coefficient exhibits a trend of initially increasing and then decreasing, while the MAE first decreases and then increases. This result is reasonable because when the G-set contains a small number of examples, it cannot fully capture the prediction consistency relationships between the source model set and the target model set. This limitation hinders the selection of the most appropriate N-set for each target model, leading to a decline in performance. As the number of examples in the G-set increases, this issue is gradually alleviated. However, when the G-set contains too many examples, as illustrated in Figure 1, the G-set may not be the most representative in the new feature space. Moreover, since the N-set includes few points beyond those from the G-set, the N-set constructed for each target model becomes less representative, resulting in a decrease in performance.

**Element-Wise Distance Effectively Facilitates Handling Various Data Forms.** Our method utilizes Element-Wise Distance, specifically the manhattan distance, enabling it to process both continuous and discrete values effectively. As shown in Table 3, the performance of the AnchorPoints method on the GSM8K dataset is substantially lower than that of our method and even falls below that of the RANDOM method. We further conduct the experiment on the distance in Table 7 under the inference count of 30, showing that the Element-wise Distance, like cosine similarity and Manhattan distance, can outperform the correlation distance on all benchmarks.

Table 7: Performance Evaluation Across Different Distance Measures.

| Distance | ARC-Challenge | | Hellaswag | | GSM8k | | Winogrande | | POPE | |
|---|---|---|---|---|---|---|---|---|---|---|
| | $\tau \uparrow$ | **MAE** $\downarrow$ | $\tau \uparrow$ | **MAE** $\downarrow$ | $\tau \uparrow$ | **MAE** $\downarrow$ | $\tau \uparrow$ | **MAE** $\downarrow$ | $\tau \uparrow$ | **MAE** $\downarrow$ |
| CORRELATION | 0.745 | 0.031 | 0.895 | 0.022 | 0.819 | 0.051 | 0.519 | 0.072 | 0.511 | 0.053 |
| COSINE | 0.740 | 0.032 | **0.911** | 0.021 | 0.825 | 0.044 | **0.610** | 0.030 | **0.562** | **0.034** |
| MANHATTAN | **0.748** | **0.026** | 0.910 | **0.017** | **0.862** | **0.036** | 0.588 | **0.028** | 0.531 | 0.043 |

**Performance with Larger Inference Count On the HellaSwag Benchmark.** To further explore the capabilities of our method when utilizing more examples, we conducted additional experiments on the HellaSwag benchmark using the first 6,000 examples and more inference count. As shown in Table 8, when the inference count is increased to 50, 100, and up to 150, our method more effectively ranks the performance of the target models and accurately reconstructs their capabilities.

Table 8: Performance of our method on the Hellaswag benchmark with the higher inference count.

| Inference Count | 50 | | 100 | | 150 | |
|---|---|---|---|---|---|---|
| | $\tau \uparrow$ | **MAE** $\downarrow$ | $\tau \uparrow$ | **MAE** $\downarrow$ | $\tau \uparrow$ | **MAE** $\downarrow$ |
| RANDOM | 0.887 | 0.053 | 0.920 | 0.038 | 0.935 | 0.030 |
| ANCHORPOINTS | 0.915 | 0.046 | 0.931 | 0.040 | 0.940 | 0.040 |
| GP-IRT | 0.869 | 0.026 | 0.915 | 0.015 | 0.936 | 0.012 |
| TAILOREDBENCH | **0.919** | **0.016** | **0.934** | **0.013** | **0.943** | **0.011** |

## 4 RELATED WORKS

**Models Correlation in Predictive Consistency:** Prior works (Miller et al., 2021; Awadalla et al., 2022; Liang et al., 2022b) have demonstrated a certain level of correlation between in-distribution (ID) and out-of-distribution (OOD) performances across diverse models and tasks. Building on this foundation, Baek et al. (2022) and Mehra et al. (2024) advance this relationship by showing the phenomenon that the agreement between two models on ID data is linearly correlated with their agreement on OOD data, where the accuracy holds the similar linear relationship, enabling accurate estimation of model's OOD accuracy based solely on ID data. Our work extends this phenomenon to address the challenge of benchmark compression, enabling the selection of more representative subsets for benchmarks.

**Coreset Selection for Efficient Benchmarking:** As LLMs proliferate and version updates accelerate, the cost of thoroughly evaluating each model across all benchmarks has become prohibitive, leading to methods that

subsample the most representative subsets from each benchmark for more efficient evaluation. Vivek et al. (2024) clusters examples directly using the confidence scores provided by source models, leveraging these scores to select an optimal subset. Similarly, Polo et al. (2024) employs an Item Response Theory (IRT) model, trained on the success matrix of each source model across various examples, to derive the latent representations of examples for clustering. Perlitz et al. (2023) proposes Flash-HELM, which dynamically adjusts the sizes of randomly selected subsets based on model ranking, where higher-ranked models are evaluated with greater precision. Prabhu et al. (2024) proposes the Sort & Search (S&S) strategy, which leverages the difficulties of examples and dynamic programming to select the coreset. Xu et al. (2024) aggregates all methods above and dynamically chooses the optimal subset selection method for each benchmark but requires many examples to determine the best approach. Despite these advancements, these methods often struggle with substantial distribution shifts between the source and target models, caused by the discrepancy between their predictive consistency, potentially causing significant distortion in estimating the target model's performance. Extending the approach of Vivek et al. (2024), our work alleviates this issue by dynamically selecting a native source model set with the highest prediction consistency to the target model, ensuring the selection of a tailored coreset for each target model that best represents the benchmark.

**Scaling Approaches for Model Performance Estimations:** Scaling law describes the relationship between model properties (e.g., FLOPs used during training, model parameter size) and model capabilities. Recent works (Hu et al., 2023; Ruan et al., 2024; Isik et al., 2024) have leveraged scaling laws to predict model performance on various downstream tasks, reducing the computational cost of evaluating models on complex downstream tasks. Zhang et al. (2024) simplifies those approaches by utilizing the relationships between model families and their collaborative overall performance across tasks rather than fitting scaling laws. The aforementioned methods typically rely on overall model performance across several benchmarks and specific design factors (e.g., model size or training data properties) to either fit scaling curves or investigate correlations between models on various tasks. In contrast, our approach addresses a more general case by reducing the evaluation cost for multiple models on a single benchmark, offering a more efficient performance estimation framework.

## 5 CONCLUSIONS

In this paper, we propose the TAILOREDBENCH method, which mainly includes an adaptive source model set selection strategy, a scalable K-Medoids clustering algorithm and a calibrated restoration strategy. Abandoning the one-size-fits-all approach, we have customized the evaluation on the constructed native coreset for each target model. This approach enables a more accurate reconstruction and ranking of the model's performance on the benchmark. Comprehensive experiments show that TAILOREDBENCH can achieve more accurate model evaluation (an average of 24.8% estimation MAE loss degradation) with only twenty inference costs.

**Limitations:** A primary limitation of mainstream approaches in benchmark compression, including Vivek et al. (2024), Polo et al. (2024), and our method, is their dependence on comprehensive evaluation results from existing models across all examples within a benchmark. As described above, these results are typically readily accessible through public leaderboards. However, for new benchmarks or certain private benchmarks, obtaining initial model performance results is necessary, which introduces additional inference overhead. Nonetheless, we maintain that this initial cost is justified, as it is offset by the significant resource savings achieved through numerous subsequent rapid evaluations facilitated by our method.

**Future Work:** An important direction for future work involves the dynamic determination of the optimal sizes for the G-set and N-set. While we have demonstrated that our method performs robustly across a variety of G-set and N-set configurations, the ability to ascertain the most suitable sizes for a specific benchmark and its corresponding source models would be highly beneficial. This advancement would enable users to better plan their computational resources in advance, optimizing the balance between inference cost and performance.

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

# A DEMONSTRATION OF METHOD EFFECTIVENESS WITH VARIANCE

In this section, we present visual comparisons of our method and other approaches, including their respective variances, as illustrated in Figures 5 through 9. The results demonstrate that our method outperforms the baseline methods on all datasets and exhibits greater robustness (with smaller variance).

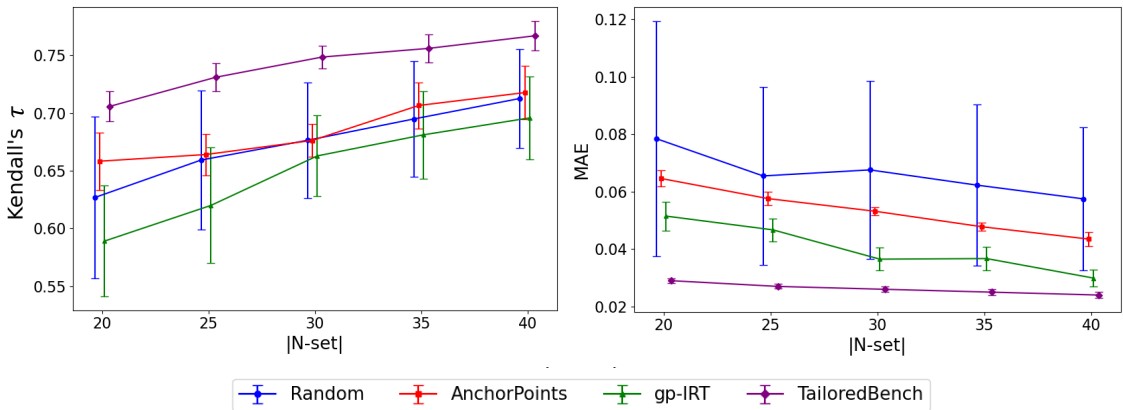

Figure 5: Demonstration of Method Effectiveness with Variance on ARC-Challenge Benchmark.

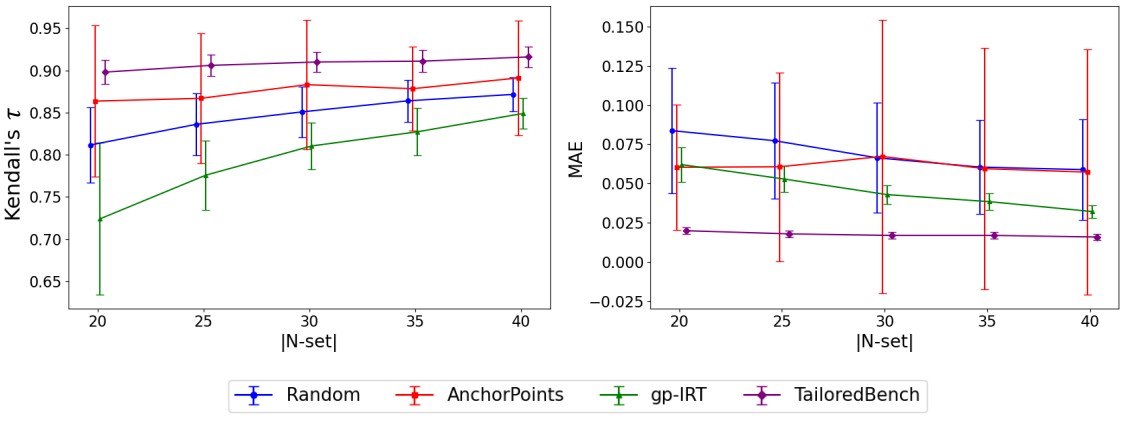

Figure 6: Demonstration of Method Effectiveness with Variance on Hellaswag Benchmark.

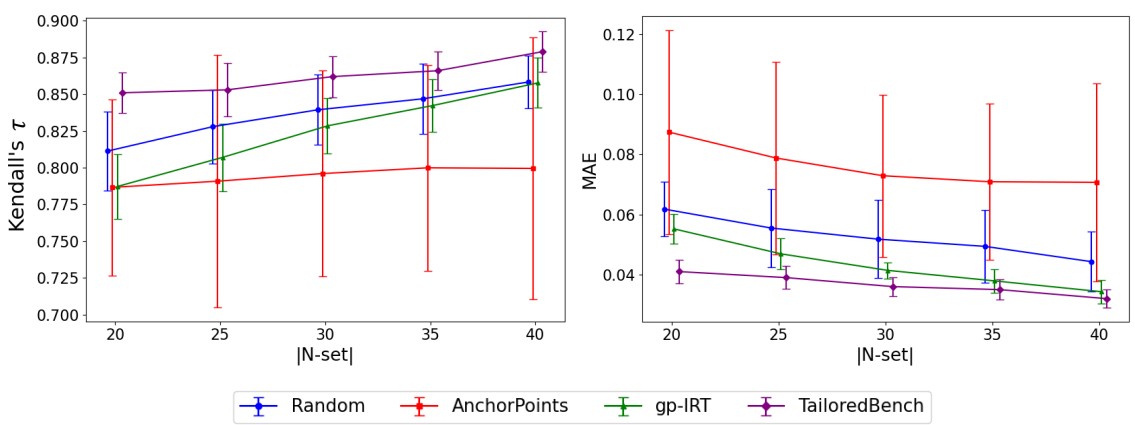

Figure 7: Demonstration of Method Effectiveness with Variance on GSM8k Benchmark.

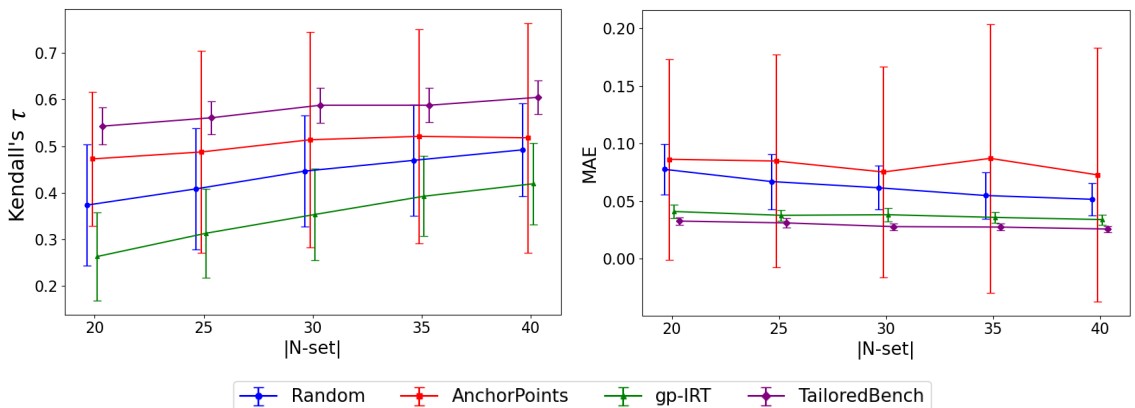

Figure 8: Demonstration of Method Effectiveness with Variance on Winogrande Benchmark.

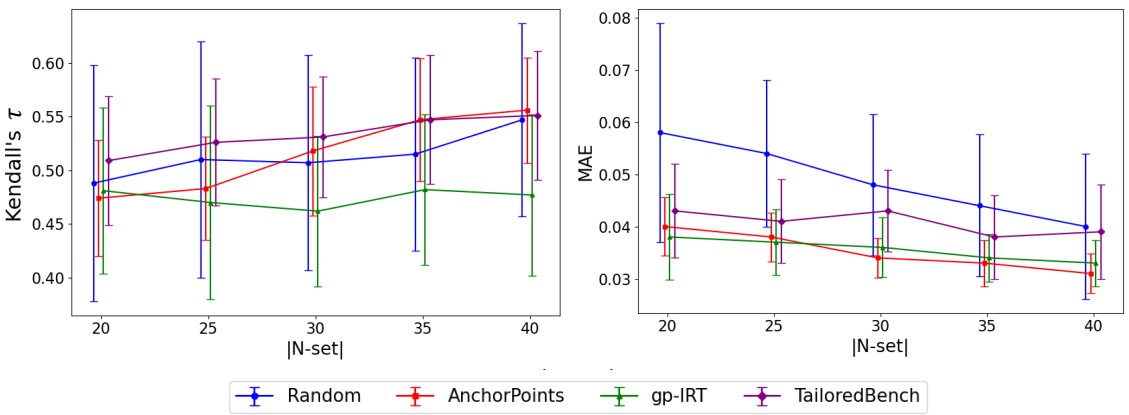

Figure 9: Demonstration of Method Effectiveness with Variance on POPE Benchmark.

# B  MORE ANALYSIS ON THE IMPACT OF NATIVE SOURCE MODEL QUANTITY ON OUR METHOD

In this section, we maintain the overall prediction consistency between the Native source models and the target models constant, while varying the proportion of the source models designated as Native Source Models from 20% to 100% for the target models across various benchmarks. The results are illustrated in Figures 10 to 14, indicating that, under the condition of maintaining the prediction consistency between the Native Source Models and the target model, the number of Native Source Models significantly influences the method's performance.

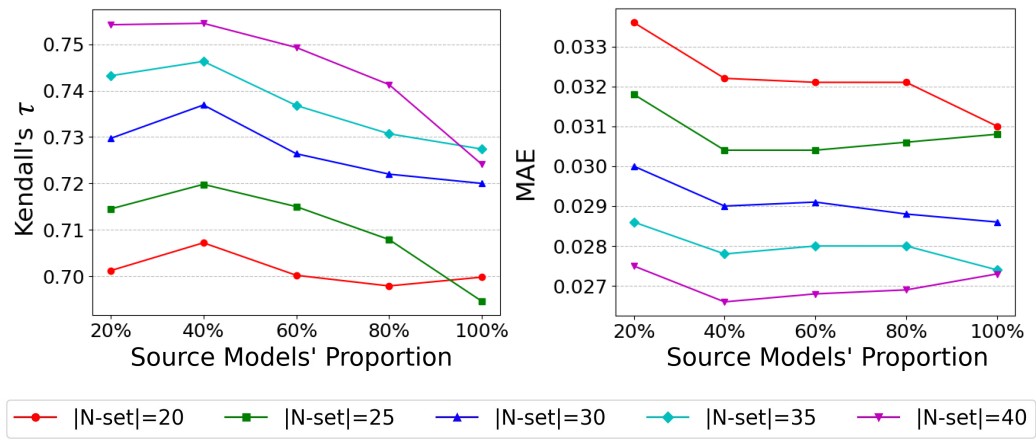

Figure 10: Effect of Native Source Model Quantity in Our Method on ARC-Challenge Benchmark.

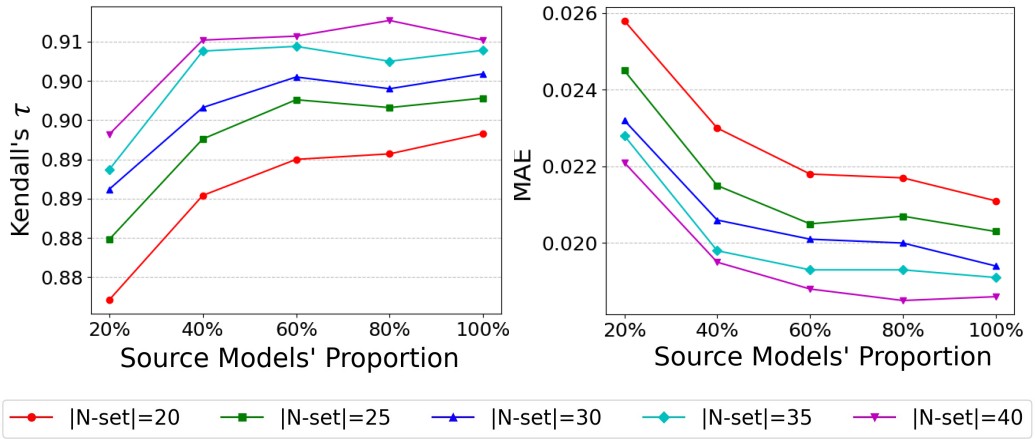

Figure 11: Effect of Native Source Model Quantity in Our Method on Hellaswag Benchmark.

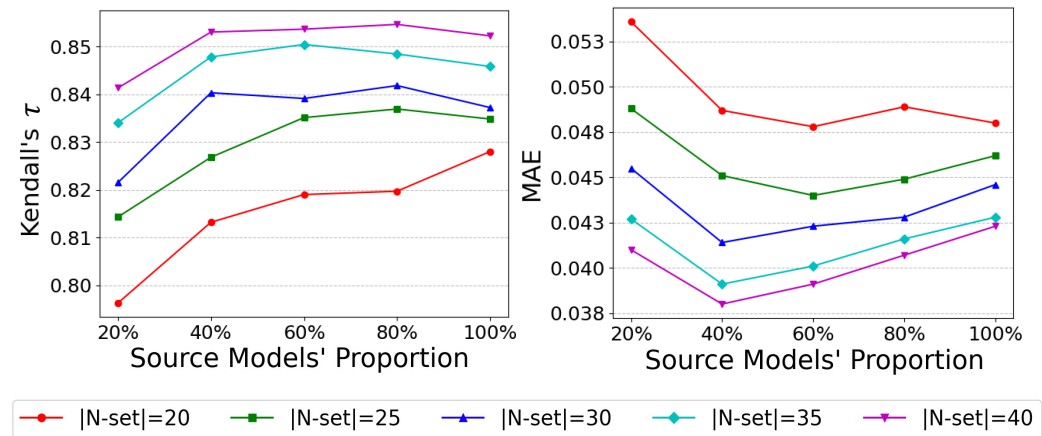

Figure 12: Effect of Native Source Model Quantity in Our Method on GSM8k Benchmark.

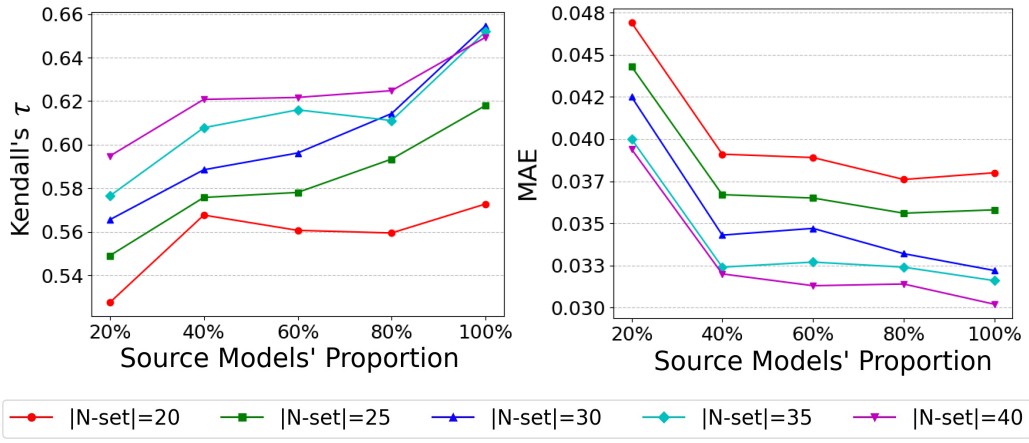

Figure 13: Effect of Native Source Model Quantity in Our Method on Winogrande Benchmark.

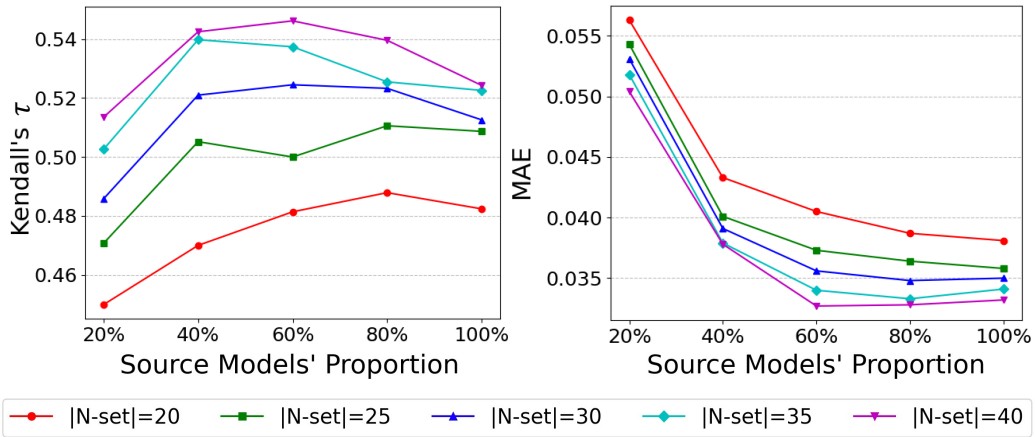

Figure 14: Effect of Native Source Model Quantity in Our Method on POPE Benchmark.

## C  MORE ANALYSIS ON THE IMPACT OF NATIVE SOURCE MODELS' PREDICTION CONSISTENCY ON OUR METHOD

We demonstrate the performance of our method by selecting Native Source Models based on their prediction consistency relative to the target model on various benchmarks, ranging from the top 20% to 80%~100%. The results are illustrated in Figures 15 to 19. The results present that, under the condition of a fixed number of Native Source Models, the performance of the method significantly decreases as the prediction consistency between the Native Source Models and the target model decreases.

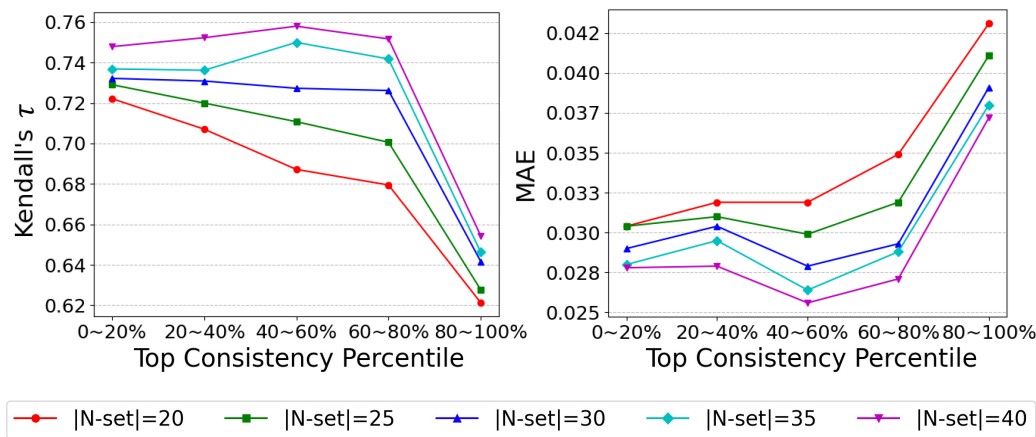

Figure 15: Effect of Native Source Models' Prediction Consistency in Our Method on ARC-Challenge Benchmark.

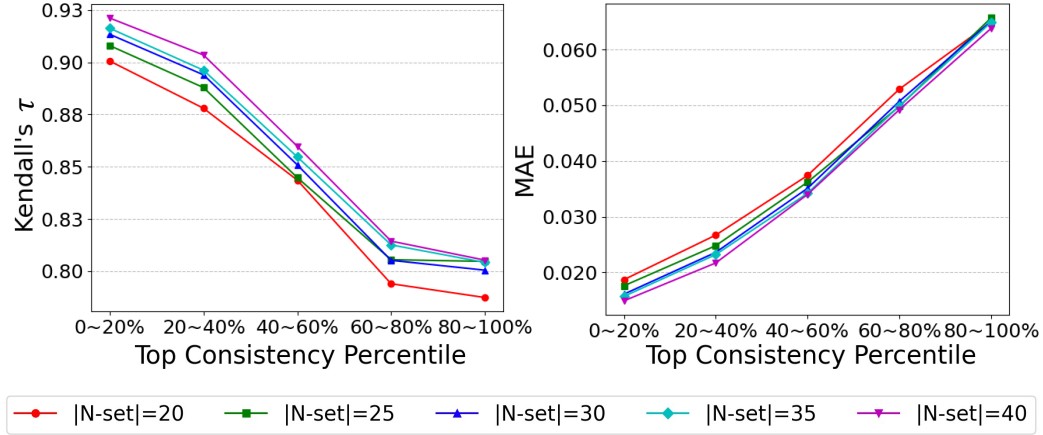

Figure 16: Effect of Native Source Models' Prediction Consistency in Our Method on Hellaswag Benchmark.

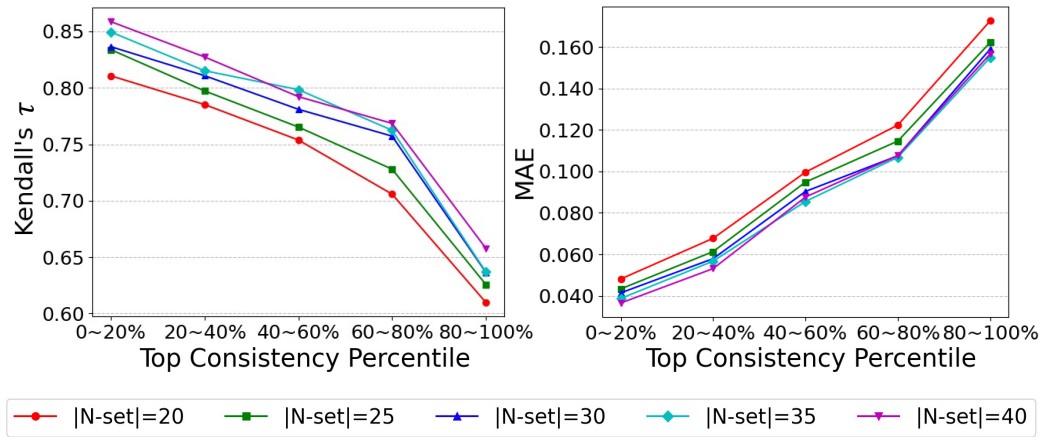

Figure 17: Effect of Native Source Models' Prediction Consistency in Our Method on GSM8k Benchmark.

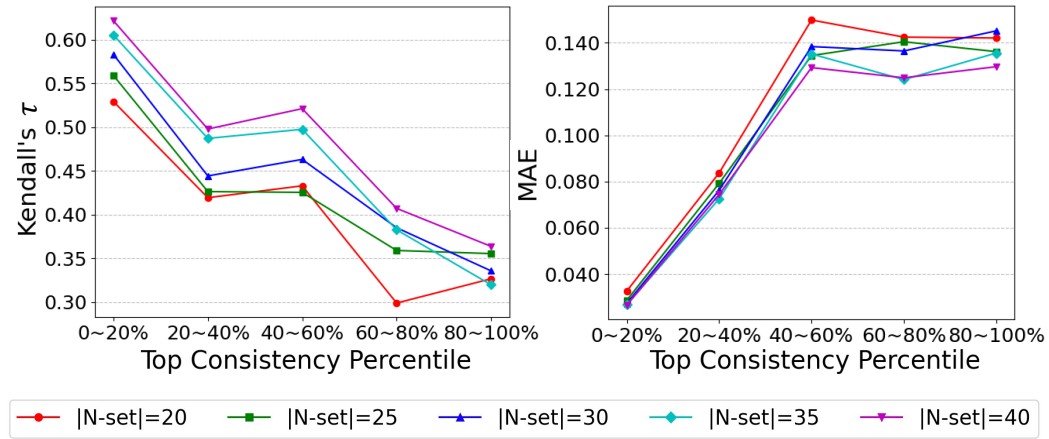

Figure 18: Effect of Native Source Models' Prediction Consistency in Our Method on Winogrande Benchmark.

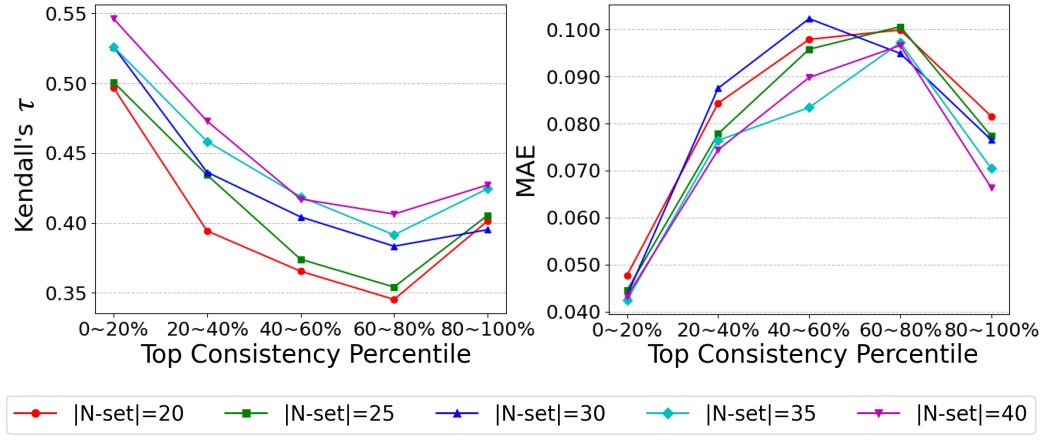

Figure 19: Effect of Native Source Models' Prediction Consistency in Our Method on POPE Benchmark.

# D    FURTHER ANALYSIS ON THE IMPACT OF THE NUMBER OF NATIVE SOURCE MODELS ON OUR METHOD

This section presents the results of our method as the number of Native Source Models is incrementally increased based on their prediction consistency with the target model. The results in Figures 20 to 23 show that, overall, Kendall's $\tau$ initially increases and then decreases as the number of Native Source Models increases, while the MAE initially decreases and then increases with the increase in the number of Native Source Models.

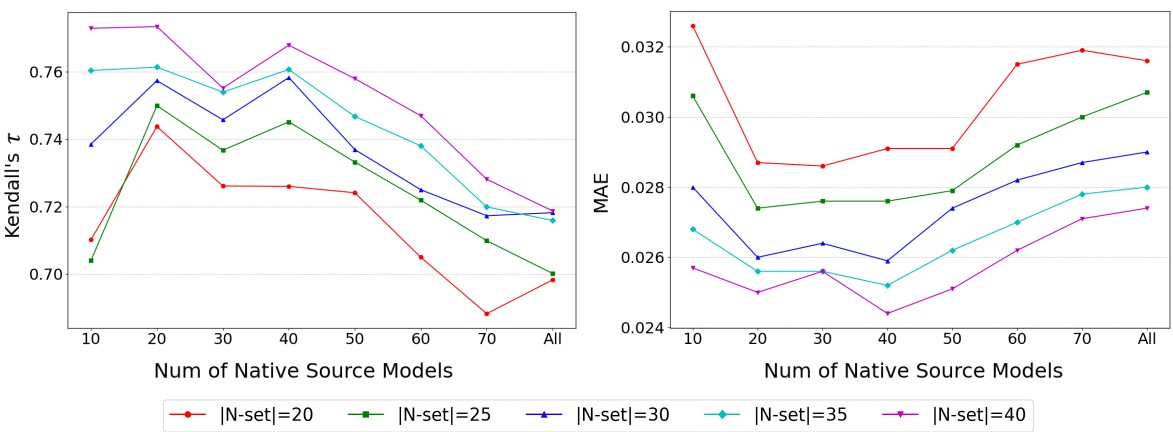

Figure 20: Effect of the Number of Native Source Models in Our Method on ARC-Challenge Benchmark.

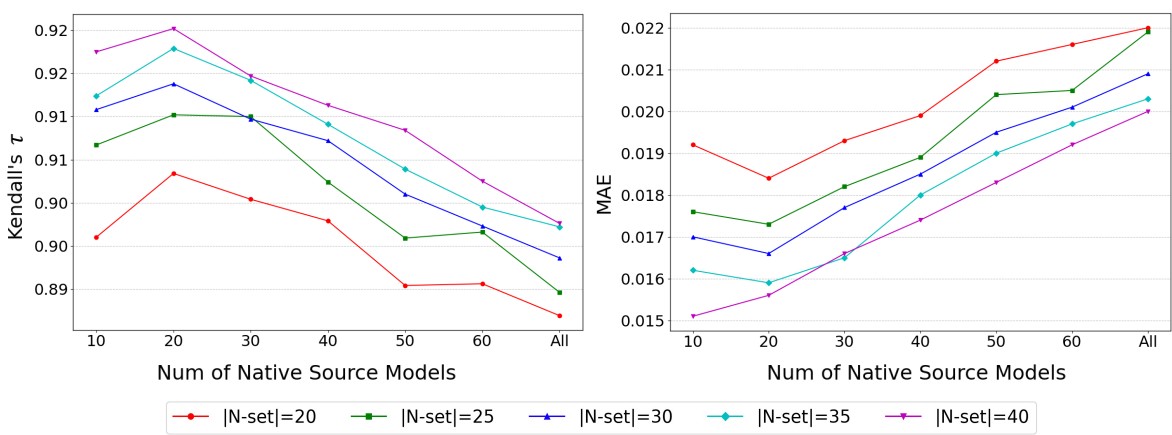

Figure 21: Effect of the Number of Native Source Models in Our Method on Hellaswag Benchmark.

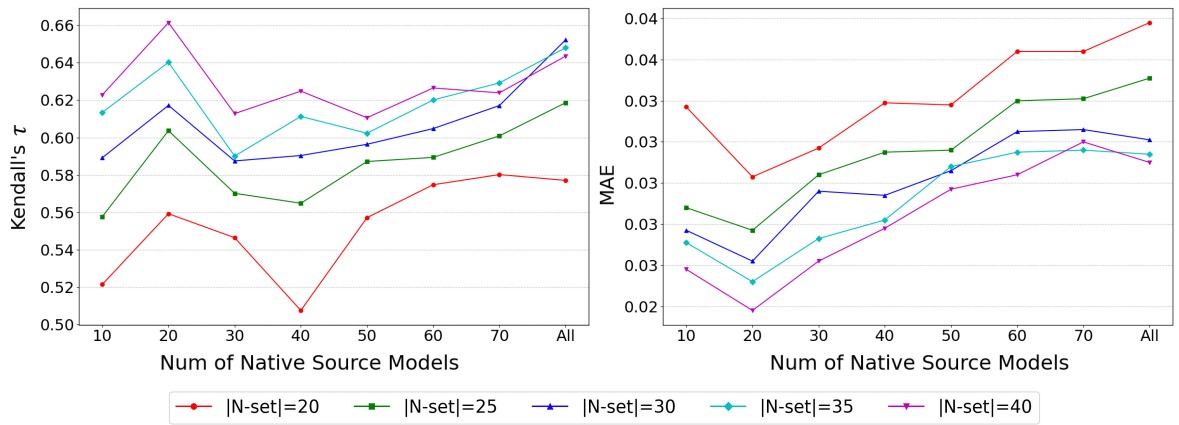

Figure 22: Effect of the Number of Native Source Models in Our Method on Winogrande Benchmark.

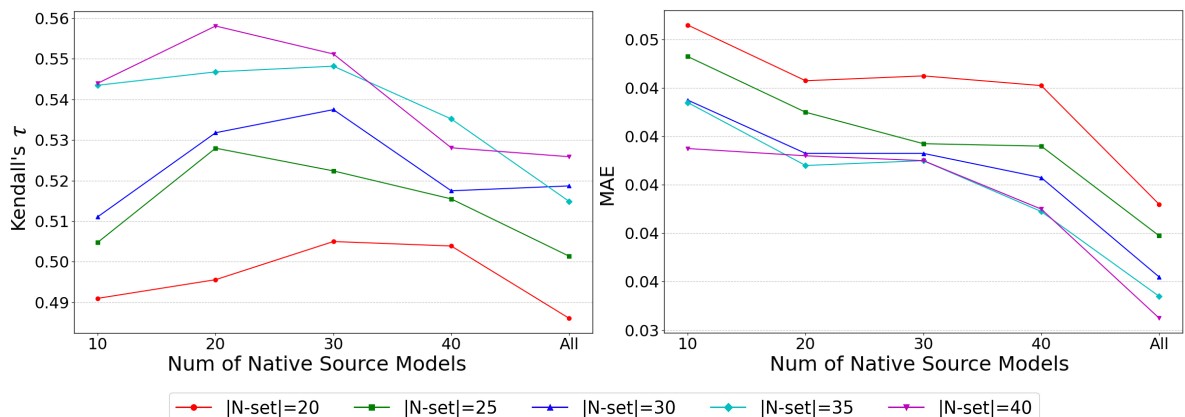

Figure 23: Effect of the Number of Native Source Models in Our Method on POPE Benchmark.

