# OpenReview forum: "Beyond One-Size-Fits-All: Tailored Benchmarks for Efficient Model Evaluation"
_ICLR.cc/2025/Conference — ICLR 2025 Conference Withdrawn Submission_

### Official Review · Reviewer_8VuP · 2024-11-01

**Soundness:** 3
**Presentation:** 3
**Contribution:** 3
**Rating:** 5
**Confidence:** 3

**Summary:**

This paper proposes a method for estimating full-data benchmark data using only limited data in order to reduce the computational cost of increasing model size in LLM benchmarking. In order to improve the imbalance in evaluation, we are conducting estimations that take into account the similarity of the model and the similarity of the data.

**Strengths:**

As various LLM models are proposed and their sizes increase, it is very important to choose the most appropriate model for your purpose at the lowest cost.  This paper proposes a method that has the potential to solve this important issue, and the results of the experiments show that it is highly effective.

**Weaknesses:**

I think that realistic use involves choosing an LLM that delivers high performance, so it is important to choose one with a high benchmark rating. On the other hand, the evaluation of the proposed method in the experiment is a comprehensive overview. To put it in the most extreme terms, in many cases, if you know that a model has low performance, it is not necessary for the figures to be accurate. It is impossible to deny the possibility that this evaluation is accurately predicting models with low performance and inaccurately predicting models with high performance.

**Questions:**

For example, is it possible to determine the accuracy of the selection of top-N models and the accuracy of prediction for each data set? If we can confirm that these evaluations are high, I think we can show that they are also valuable from a practical point of view. I would like to evaluate the value of the final method by taking these results into account.

---

### Official Review · Reviewer_AyN2 · 2024-11-04

**Soundness:** 3
**Presentation:** 2
**Contribution:** 2
**Rating:** 3
**Confidence:** 4

**Summary:**

Scaling up large models has enhanced their capabilities but poses challenges for efficient evaluation, as testing billion-parameter models can be costly in terms of GPU hours and API use. Existing evaluation methods use a static coreset based on source models’ predictions to approximate a target model's performance, assuming prediction consistency across models. However, this assumption can fail in practice, as target models do not always exhibit similar prediction patterns, leading to inaccurate performance estimates. To address this, the authors propose TAILOREDBENCH that builds a dynamic, model-specific evaluation set by combining a static global coreset with a model-specific subset (N-set) for each target model. Using empirical studies, the authors show that this tailored approach can enable more accurate evaluations compared to existing methods.

**Strengths:**

+ The paper studies a practical problem of evaluating large-scale models with limited inference budgets. At its core, this problem is formulated as coreset construction. The key novelty is in customizing the coreset construction for each target model, as opposed to identifying a global set of samples for all target models.
+ Though the proposed algorithm is a straightforward extension of ANCHORPOINTS (K-Mediods clustering), the idea of sub-selecting source models for prediction consistency is interesting.

**Weaknesses:**

The paper is not clearly written and is very difficult to follow.
+ What do the terms prior and posterior mean in the context of model evaluation and coreset selection? The terms are used hand-wavily without sufficient clarity
+ The notations are not fully consistent. The paper needs a thorough proof-read.
+ The experimental setup is not clear. For each dataset, the authors report that <xx> number of models were chosen. However, no details are provided on the actual evaluation setup. It would be helpful for improving the reproducibility of this work.
+ What does a distribution shift between source and target model imply? The authors need to formally define the assumptions. If all models are chosen from a common leaderboard, it is not guaranteed that all source models were trained on the same data. In that case, what does distribution shift mean?

The method itself is fairly simple, which is not a negative in my opinion. However, even simpler methods require proper justification to understand its behavior. In its current presentation, it is very difficult to fully comprehend the design choices made.
+ For example, even though the authors pick source models from leaderboards, getting the predictions for all data samples from the set of source models is computationally expensive. Why is that cheaper than evaluating a set of target models?

Though I am providing a lower rating to the current submission, I am open to improving my score based on the updated manuscript during the rebuttal phase.

**Questions:**

see weaknesses

---

### Official Review · Reviewer_zVeh · 2024-11-06

**Soundness:** 3
**Presentation:** 1
**Contribution:** 2
**Rating:** 3
**Confidence:** 3

**Summary:**

Evaluating LLM on heterogeneous benchmarks is very costly. The authors propose a method to reduce the cost by using a clustering algorithm (K-Medoids) to select a representative subset of examples in the benchmarks. The core inovation proposed in this paper is to add to this representative subset different examples for each model to be evaluated -- aka target models. These extra examples provides a better estimation of the model's performance, as they are tailored to the specific model. The authors show that this method is more efficient than other existing methods on a benchmark with 5 tasks and aroudn 150 models.

**Strengths:**

- **S1:** The topic is timely and well justified.
- **S2:** The method effectively reduces the cost of evaluating LLM while preserving the heterogeneity of the tasks.

**Weaknesses:**

- **W1:** The paper is not clear. Many quantities and concepts are vaguely introduced, making the paper hard to follow, in particular when not familiar with LLM evaluation methods. For instance, the concept of corectness of the model is not defined. I guess this can be any type of loss that evaluate how good $\phi(x_k)$ is compared to $y_k$. But this should be explicit. Also, indicating the dimension of the vectors -- like $\dot x_k, \dot \phi_n$ -- would help to understand the paper.
- **W2:** The paper is of limited novelty and does not properly put in context the existing work and the novel contributions. For instance, section 2.2 is exactly the same as section 4 in Vivek et a. 2024, and the `CALIBRATED RESTORATION` strategy seems to correspond to the Anchor Point Weighted in the same paper. The core innovation of the paper is to add the `Adaptive native source model`, and this should be clear that the rest is existing work.
- **W3:** The evaluation procedure is not clear (see **Q1** and **Q2** ). This draws questions about the usefulness of the method to evaluate novel models that are completely unrelated to the source models.
- **W4:** From Table 1 to 5 and Table 8, it seems that TailoredBench has even larger diminishing returns than the ones reported in AnchorPoints. For instance on HellaSwag, N-150 is only twice as good ad N=20. This could indicate a larger overfit/bias due to not considering the generalization properties of the model.

**Questions:**

- **Q1:** How are the individual model performance evaluated? It is not clear from the paper. In particular, it would be quite important to have a test set to evaluate the performances of the model that is not used when computing the clustering. Otherwise, this procedure would be biased and not estimate the true generalization performances of the model, which is ultimately what we want to estimate.
- **Q2:** what is the list of model that are used for the evaluation and how does the split between source and target account for the fact that some model are related? (same model family, same training, ... ).



### Minor comments, nitpicks and typos

- l.146 -> $y_k$ is introduced but never used.
- l.157 -> `and the superscript $\mathcal S$ indicates the model set that providing the correctness for each example, comprising the dimension of the example vector.` -> This sentence is not clear. Is it the `model set that provides the correctness for each example`? What is the `dimension of the example vector`?
- Eq (1) -> The K-medoid also optimizes for $\mathcal C_g$.
- l.173: `the cluster whose centroid is $g$.` should be $x_g$.
- l.223: why not try greedy initialization, as it is common in K-means? This usually improves the stability of the algorithm.

---

### Note · Authors · 2024-11-21

**Comment:**

Dear Reviewers,

We sincerely appreciate the time and effort you have devoted to reviewing our paper.

We are grateful to reviewers zVeh and 8VuP for recognizing the timeliness and significance of our proposed method.

Overall, we agree with the reviewers' observation that our writing may have caused some misunderstandings, particularly for reviewers zVeh and AyN2. For this, we offer our apologies.

However, we respectfully disagree with the assertion by reviewers zVeh and AyN2 that our method lacks novelty. Specifically, taking into account the diverse scoring formats of different models, our approach adopts item-based distance in the "Constructing G-set" component, which distinguishes it from prior works and is extensively discussed in our manuscript. Furthermore, our primary contributions lie in the "Adaptive Native Source Model Selection" and "Developing N-set" sections, which are fundamentally distinct from existing methods and significantly enhance the effectiveness of our approach.

Your valuable comments have provided us with insights to further refine our work for future submissions.

Best regards

**Withdrawal Confirmation:**

I have read and agree with the venue's withdrawal policy on behalf of myself and my co-authors.